# Early life stress alters transcriptomic patterning across reward circuitry in male and female mice

Catherine Jensen Peña [1,2]*, Milo Smith[1,3], Aarthi Ramakrishnan[1], Hannah M. Cates [1], Rosemary C. Bagot[1,4], Hope G. Kronman[1], Bhakti Patel[1], Austin B. Chang[2], Immanuel Purushothaman[1], Joel Dudley[1,3], Hirofumi Morishita [1], Li Shen [1] & Eric J. Nestler [1]*

Abuse, neglect, and other forms of early life stress (ELS) significantly increase risk for psychiatric disorders including depression. In this study, we show that ELS in a postnatal sensitive period increases sensitivity to adult stress in female mice, consistent with our earlier findings in male mice. We used RNA-sequencing in the ventral tegmental area, nucleus accumbens, and prefrontal cortex of male and female mice to show that adult stress is distinctly represented in the brain's transcriptome depending on ELS history. We identify: 1) biological pathways disrupted after ELS and associated with increased behavioral stress sensitivity, 2) putative transcriptional regulators of the effect of ELS on adult stress response, and 3) subsets of primed genes specifically associated with latent behavioral changes. We also provide transcriptomic evidence that ELS increases sensitivity to future stress through enhancement of known programs of cortical plasticity.

---

[1] Nash Family Department of Neuroscience and Friedman Brain Institute, Icahn School of Medicine at Mount Sinai, New York, NY 10029, USA. [2] Princeton Neuroscience Institute, Princeton University, Princeton, NJ 08544, USA. [3] Institute for Next Generation Healthcare, Icahn School of Medicine at Mount Sinai, New York, NY 10029, USA. [4] Present address: Department of Psychology, McGill University, Montréal, QC H3A 1B1, Canada. *email: cpena@princeton.edu; eric.nestler@mssm.edu

Early life stress (ELS) increases the lifetime risk of depression and is one of the strongest risk factors for developing mental illness[1–3]. Nearly 1% of children in the US are victims of child abuse and neglect each year[4], and many more children experience other forms of profound stress and trauma, including death of a caregiver, parental incarceration, or experience of a natural disaster. Studies in humans and animal models indicate that ELS increases risk for depression by sensitizing individuals to stress later in life, leading to a first appearance or synergistic worsening of depression-like symptoms[5–9]. Stress early in life can alter developmental brain trajectories, and not just steady-state processes, which has the potential to augment the long-term impact of stress. However, the ways in which ELS alters transcriptional development in the brain to increase risk for depression and depression-like behaviors are poorly understood.

We recently reported that ELS during a sensitive period from postnatal day P10–20, but not P2–12, increases the susceptibility to develop depression-related behavioral abnormalities after a second hit of stress in adulthood in male mice[7,8]. Here, we first demonstrate that ELS in the same postnatal period likewise increases the susceptibility of female mice to adult stress. Using RNA-sequencing, we next asked whether ELS in male and female mice alters patterns of gene transcription in the brain's reward circuitry, including ventral tegmental area (VTA), nucleus accumbens (NAc), and prefrontal cortex (PFC), all of which are implicated in stress responses and human depression. Our previous work found broad and long-lasting transcriptional changes in adult male VTA after ELS[7]. We hypothesized that similar biological pathways might be disrupted in male and female mice by ELS, even if specific genes within each pathway are distinct between the sexes. We also hypothesized that ELS-induced transcriptional changes in these brain regions would in turn lead to either unique transcriptional responses to additional stress in adulthood, or an exaggeration of earlier transcriptional alterations. We used a variety of analytical tools to determine biological processes, primed genes, and putative regulatory factors disrupted by ELS alone or in combination with adult stress to address these questions.

Overall, we present comprehensive genome-wide analysis of transcriptional changes in both male and female mice, across three brain reward regions, after ELS with or without subsequent adult stress. These transcriptional analyses provide a foundation for future research testing the causal role of molecular pathways implicated in ELS-induced stress sensitivity in both sexes.

## Results

**ELS increases susceptibility to adult stress in female mice**. We measured depression-like behaviors in female mice on four individual tests and in a composite behavior outcome score (see the "Methods" section). Adult sub-threshold variable stress (STVS) alone did not significantly alter behavior relative to standard-reared control female mice on any measure. There was an interaction between ELS and STVS on latency to eat (Fig. 1b) and on the ratio of latency to eat in a novel vs. home cage environment (Fig. 1c) in the novelty-suppressed feeding test, a test of anxiety-like behavior that is also sensitive to chronic but not acute antidepressant treatment similar to responses in human patients[10]. There were also main effects of both ELS and STVS on the ratio of novel/home cage latencies. Female mice that experienced two hits of stress had increased latency and ratios relative to ELS and STVS alone. Survival analysis indicated differences in latency to eat among ELS–STVS mice compared to ELS mice (Fig. 1d, Log-rank Mantel–Cox: $X^2 = 4.986$, $p = 0.0256$) and STVS mice (Log-rank Mantel–Cox: $X^2 = 5.252$, $p = 0.0219$). Main effects of both ELS and STVS on distance traveled (Fig. 1e)

in the NSF may differences in non-ambulatory time spent eating rather than hyperactivity per se.

In the forced-swim test[11] there was a main effect of STVS on latency to immobility, and female mice that experienced both ELS and STVS had shorter latency to immobility compared to ELS alone (Fig. 1f). There were no differences in total immobility (Fig. 1g). The two-bottle choice sucrose preference test[12] revealed no effect of either stress alone or in combination in female mice (Fig. 1h). Grooming duration after being sprayed with a 10% sucrose solution[13] showed a main effect of ELS, but no interaction with STVS, such that ELS decreased grooming duration (Fig. 1i).

We generated a composite behavior outcome score[7] of these behavioral responses and found a main effect of ELS (Fig. 1j). The combination of ELS and adult stress in female mice significantly increased this composite score relative to STVS alone, similar to our earlier findings in male mice on a different battery of behavioral tests before and after repeated social defeat stress (Fig. 1l)[7]. There was also an interaction of stress types on body weight in adulthood in female mice, such that ELS increased weight among control mice but decreased weight after STVS (Fig. 1k).

We next sought to replicate the novelty suppressed feeding, grooming, and sucrose preference findings in an independent cohort of female mice, and simultaneously test whether stress earlier in the postnatal period (P2–10, termed early–ELS) also heightens later stress sensitivity. We did not find an effect of early–ELS on any individual measure (Supplementary Fig. 1a–c). We then analyzed the replication cohort including only late ELS groups, as in the original cohort. We again found a main effect of ELS and STVS on the NSF ratio of latencies to eat in the novel arena vs. home cage (Supplementary Fig. 1a), but no effects on splash test grooming duration, sucrose preference, or the composite score in the replication cohort alone (Supplementary Fig. 1b, c).

We evaluated whether phase of estrus cycle impacted behavior in any test (for example, whether estrus reduced depression-like behavior, or failed to reduce depression-like behavior among ELS mice[14]), but did not find a significant relationship.

**Distinct transcriptional patterns depend on history of ELS**. In order to investigate the transcriptional correlates of ELS and adult stress, alone or in combination, we performed poly-A selected RNAseq from whole-tissue punches of adult male and female VTA, NAc, and PFC. Female brain samples were taken from mice behaviorally characterized in the initial cohort here (Fig. 1), while male brain samples were from mice whose behavior was reported previously[7]. An analysis of differentially expressed genes (DEGs) was previously reported for male VTA only after either ELS or social defeat alone[7]. Here we extend this analysis and include transcriptional changes after ELS, adult stress (STVS or Defeat), and a combination of the two stresses (ELS + STVS/Defeat), in three brain reward regions from adult males and females. Significance of DEGs was set at uncorrected $p < 0.05$ and $\log_2$(fold-change (FC)) > |0.3785| (LFC, corresponding to FC > 30%) for broad pattern identification. There was 1–23% overlap in DEGs across the three stress conditions (ELS or STVS/Defeat or ELS + STVS/Defeat) in male and female VTA, male NAc, and male and female PFC (Fig. 2a, f, i, q, v), and more than 50% overlap in female NAc, compared to a standard-reared control with no adult stress (Std-Ctl; Fig. 2n).

Although the focus of the current analysis is on the broad patterns of change induced by stress, we attempted to validate a subset of individual DEGs from the ELS vs. Std comparison from each sex and brain region using a completely independent cohort of animals. While not all differences reached statistical

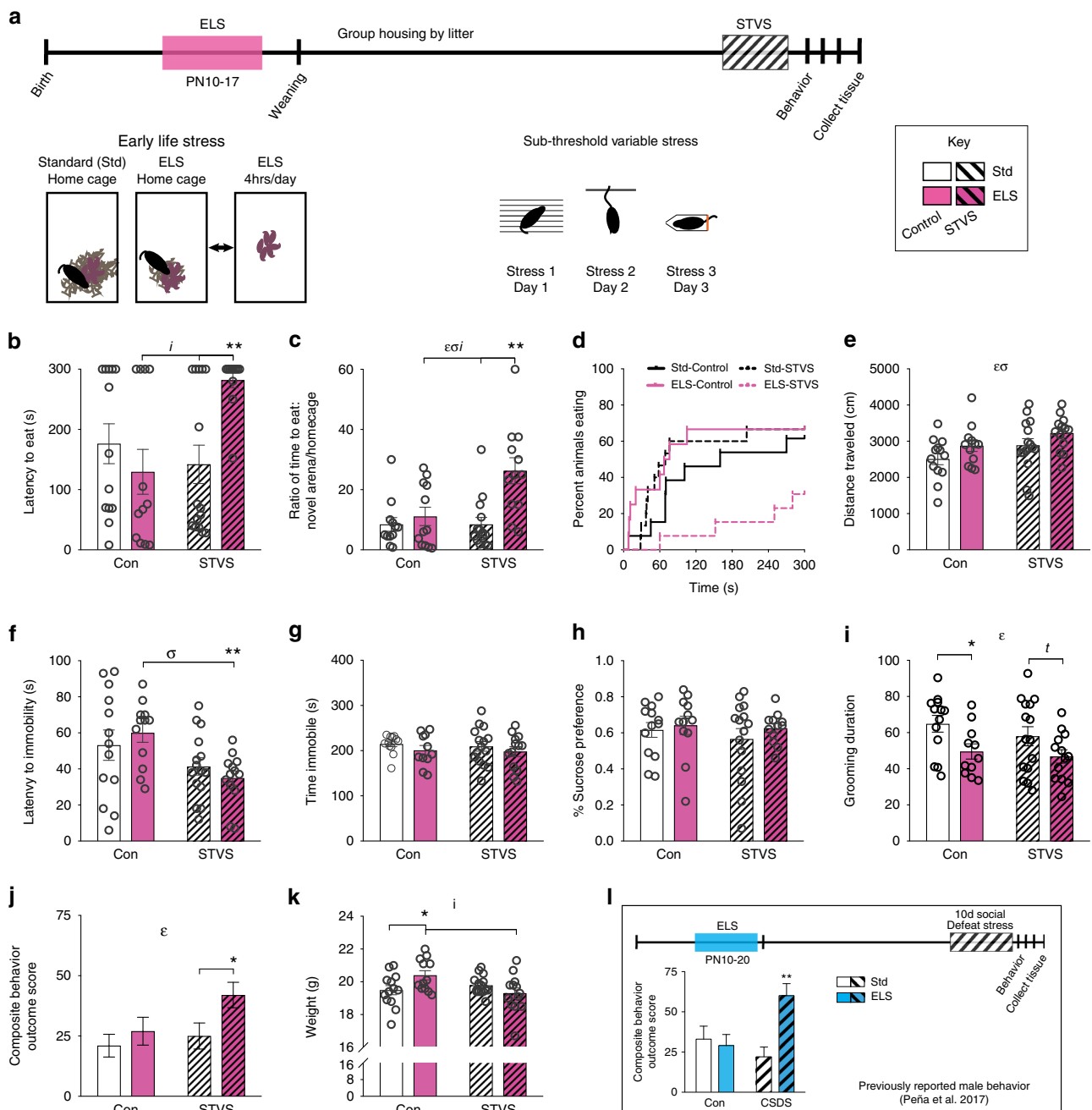

**Fig. 1** ELS increases depression-like behavior in female mice after adult STVS. **a** Schematic timeline and stress paradigms in female mice. **b** Latency to eat in a novel arena (NSF) was affected by both ELS and STVS (interaction in two-way ANOVA: $F_{1,48} = 9.158$, $p = 0.004$). ELS + STVS latencies were longer compared to ELS (two-sided $t$-test: $t_{1,22} = 3.865$, $p = 0.001$) and STVS alone ($t_{1,25} = 3.744$, $p = 0.001$). **c** Ratio of latencies to eat in a novel area/home cage was affected by ELS and STVS (interaction: $F_{1,46} = 6.119$, $p = 0.017$); ELS main effect: ($F_{1,46} = 6.169$, $p = 0.017$); STVS main effect: ($F_{1,46} = 11.201$, $p = 0.002$). ELS + STVS ratios were elevated compared to ELS ($t_{1,22} = 2.866$, $p = 0.009$) and STVS ($t_{1,24} = 3.766$, $p = 0.001$). **d** Survival plot of NSF latencies. **e** NSF distance traveled was affected by both ELS (main effect: $F_{1,48} = 4.607$, $p = 0.037$) and STVS (main effect: $F_{1,48} = 4.885$, $p = 0.032$). **f** Latency to immobility in a forced-swim test was affected by STVS (main effect: $F_{1,49} = 10.049$, $p = 0.003$) and was decreased by ELS + STVS relative to ELS alone (exploratory $t$-test: $t_{1,23} = 3.884$, $p = 0.001$). **g** Total time immobile in a forced-swim test and **h** sucrose preference were unaffected by either stress in female mice. **i** Grooming duration in a splash test was reduced by ELS (main effect: $F_{1,48} = 8.156$, $p = 0.006$). **j** A composite behavior outcome score was elevated by ELS (main effect: $F_{1,49} = 4.742$, $p = 0.034$) with a trend for a main effect of STVS ($F_{1,49} = 3.194$, $p = 0.080$). ELS + STVS increased this score relative to STVS alone (exploratory two-sided $t$-test: $t_{1,26} = 2.275$, $p = 0.031$). **k** Adult weight was impacted by both ELS and STVS (interaction: $F_{1,49} = 7.212$, $p = 0.010$). ELS females weighed more than standard control ($t_{1,23} = 2.284$, $p = 0.032$) and ELS + STVS ($t_{1,23} = 2.558$, $p = 0.018$). **l** Male mouse paradigm schematic and composite behavior outcome score for comparison (adapted from[7]). Std = standard-reared, Con = control (no adult stress). Error bars indicate mean ± SEM. Significant ($p < 0.05$) two-way ANOVA interaction ($i$), main effect of ELS ($\varepsilon$), and main effect of adult stress ($\sigma$) as indicated. For direct comparison of two groups: **$p < 0.01$, *$p < 0.05$, $t$ $p < 0.1$. Source data is available as a Source Data file

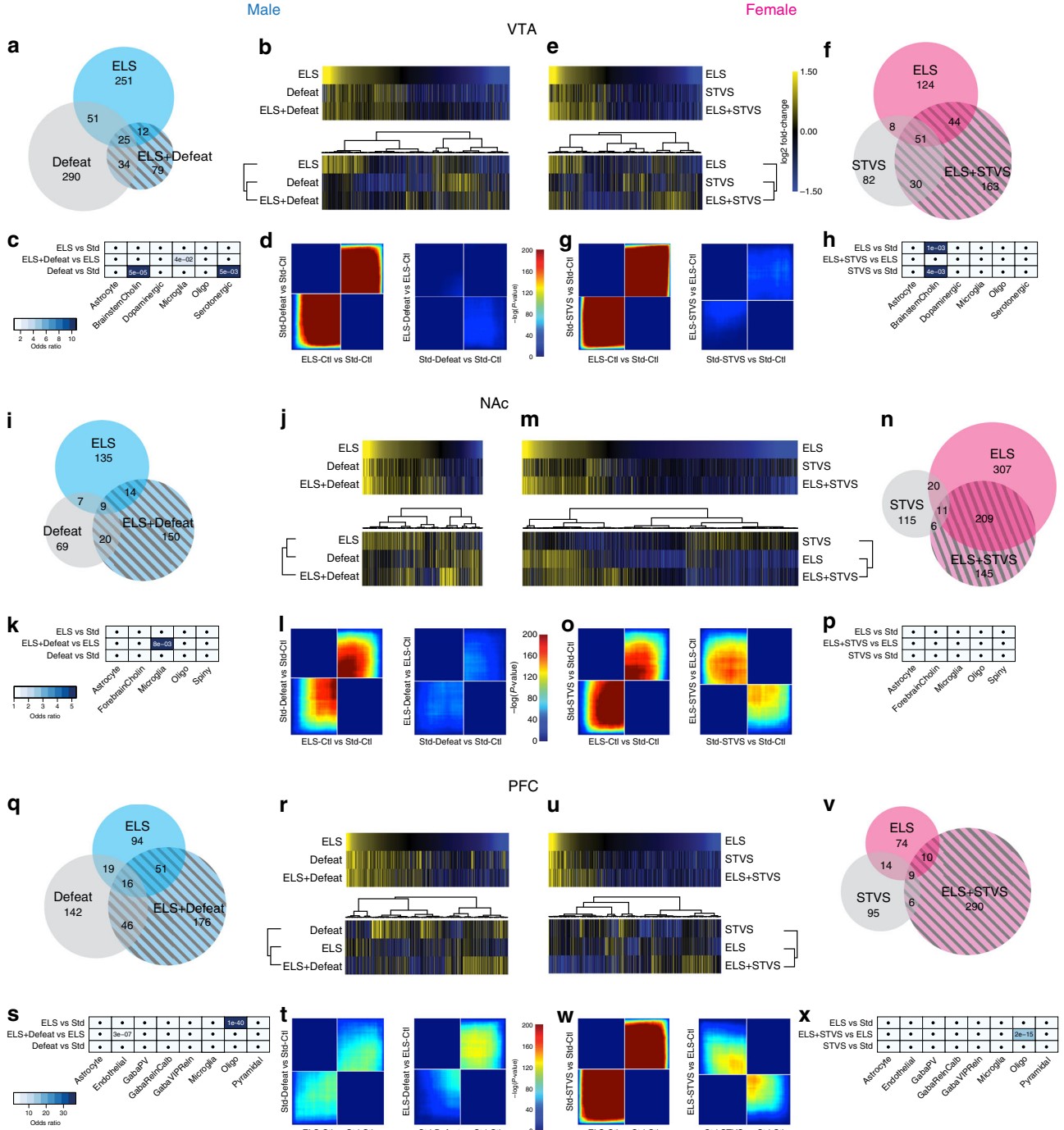

**Fig. 2** ELS alters broad transcriptional patterns within male and female brain reward regions. DEGs in adult male VTA **a–d**, adult female VTA **e–h**, adult male NAc **i–l**, adult female NAc **m–p**, adult male PFC **q–t**, and adult female PFC **u–x**. All DEGs (uncorrected $p < 0.05$) for Venn diagrams and heatmaps are relative to standard-control of the matched sex/region. Venn diagrams represent number of DEGs **a**, **f**, **l**, **n**, **q**, **v** altered by ELS, adult stress (Defeat or STVS), or a combination of stresses within each sex and brain region. Union heatmaps of DEGs **b**, **e**, **j**, **m**, **r**, **u**, top represent LFC of genes in matched comparisons regardless of significance: yellow indicates increasing $\log_2$(fold-change) expression; blue represents decreasing expression. Clustering of these heatmaps (**b**, **e**, **j**, **m**, **r**, **u**, bottom) reveal differences between stress comparisons. Enrichment of DEGs in each comparison with cell-type specific genes found in midbrain **c**, **h**, striatum **k**, **p**, or PFC **s**, **x**. Significance of enrichment is indicated in each cell and shaded by degree of odds ratio, by region. **d**, **g**, **l**, **o**, **t**, **w** Threshold-free comparison of DEGs by rank-rank hypergeometric overlap. Pixels represent the overlap between the transcriptome of each comparison as noted, with the significance of overlap ($-\log_{10}$($p$-value) of a hypergeometric test) color coded. Lower left quadrant include co-upregulated genes, upper right quadrant include co-downregulated genes, and upper left and lower right quadrants include oppositely regulated genes (up-down and down-up, respectively). Genes along each axis are sorted from most to least significantly regulated from the middle to outer corners

significance, 72% of the genes were altered in the direction predicted from the sequencing data (Supplementary Fig. 2). This suggests that the broad patterns of transcriptional change following ELS hold across cohorts.

Union heatmaps sorted by LFC of ELS DEGs showed similarities in direction of expression change across stress conditions compared to Std-Ctl (Fig. 2b, e, j, m, r, u, top). Clustering revealed greater transcriptional similarities between two hits of stress and ELS alone in female NAc and male and female PFC (Fig. 2m, r, u, bottom, respectively), greater similarities between two hits of stress and adult stress alone in male and female VTA (Fig. 2b, e, bottom), and greater similarity between ELS and adult stress alone in male NAc (Fig. 2j, bottom).

We complemented these analyses with a two-sided rank–rank hypergeometric overlap analysis (RRHO[15]) to identify patterns and strength of genome-wide overlap in a threshold-free manner. RRHO analyses confirmed similar directional regulation of genes by ELS and adult stresses (Fig. 2d, g, l, o, w, left) in all brain regions (Fig. 2t, left). As previously reported for male VTA[7], these analyses suggest that late postnatal stress transcriptionally primes VTA to be in a depression-like state, even prior to emergence of depression-like behaviors. However, RRHO analysis revealed that adult stress after ELS induces a transcriptionally unique signature compared to adult stress after standard rearing (in all regions except male PFC; Fig. 2d, g, l, o, w, right), with a significant opposite regulation of gene expression in female NAc and PFC (Fig. 2o, w, right). In other words, the impact of adult stress was transcriptionally different depending on the history of ELS. These unique latent signatures revealed by adult stress suggest transcriptional priming or altered cellular or morphological development of these brain structures.

We next asked whether ELS, adult stress, or a combination of stresses induced cell type-specific transcriptional changes. Enrichment analysis using brain region-specific curated cell type-specific marker lists[16] revealed enrichment of brainstem cholinergic-specific genes in male and female VTA comparisons and oligodendrocyte-specific genes in male and female PFC comparisons, although the specific comparisons differed (Fig. 2c, h, k, p, s, x). Overall, however, we found little cell type specificity, indicating instead that DEGs were expressed by multiple cell types within each brain region.

**ELS alters sex-specific transcriptional regulatory pathways.** We used three complementary analyses to assess how ELS affects long-term molecular regulation within the brain's reward circuitry compared to standard-reared controls: (1) DAVID functional annotation for gene ontology of biological processes, (2) upstream regulator analysis with ingenuity pathway analysis, and (3) HOMER motif analysis to predict differential transcriptional regulators. These analyses were agnostic to direction of expression change. We hypothesized that, while distinct sets of genes may be regulated across sexes or across brain regions, similar biological processes or transcriptional regulators might be affected. Top gene ontology terms for male and female VTA, NAc, and PFC are shown in Fig. 3a, d, and g. Cell differentiation was the only biological process with altered enrichment in both males and females within one brain region, NAc (Fig. 3d). Multicellular organism development and nervous system development also had altered enrichment across brain regions and sexes, together suggesting that ELS in this sensitive window causes long-lasting alterations in development of this reward circuitry. Top predicted upstream regulators of transcriptional alterations were distinct between males and females, but alpha-synuclein (SNCA) and beta catenin (CTNNB1) were both predicted upstream regulators in female VTA and NAc (Fig. 3b, e, h). Analysis of enriched

transcription factor binding sites did not reveal commonality across brain regions or sexes. Of interest was the finding that OTX2, which we showed previously is an important upstream regulator of ELS-induced transcription in the male VTA[7], did not show this predicted role in the female VTA or in NAc or PFC in either sex.

We next examined molecular pathways altered by a second hit of stress after ELS, compared to ELS alone. Multicellular organism development was again a top GO term in male VTA and male and female PFC (Fig. 4a and g). Interestingly, the effect of a second hit of stress after ELS was predicted to be regulated by common steroid hormones and their receptors in males and females, such that testosterone was a predicted upstream regulator in male and female VTA, ESR1 (estrogen receptor alpha) in male and female NAc, and beta-estradiol in male and female VTA (Fig. 4b, e, h). While there were no common enriched transcription factor binding sites in DEG lists between males and females, a majority of the enriched factors after a second hit of stress (Fig. 4c, f, i) were the same as those enriched after ELS alone (Fig. 3c, f, i), suggesting either continued or common regulation of these factors through adult stress.

**ELS primes a subset of genes for latent stress response.** We defined primed genes as those in the ELS + adult stress group significantly ($p < 0.05$) different from standard-control expression levels with LFC > |0.3875|, and also significantly ($p < 0.05$) different from both ELS-alone and adult stress-alone groups, in either direction without FC cutoff (Fig. 5a). Primed genes were unique by brain region (Fig. 5b). Males and females shared no primed genes in VTA, 10 primed genes in NAc (*Abca4, Cldn2, Col8a1, Kcnj13, Krt18, Slc4a5, Steap1, Wdr72, Wfdc2,* 1500015O10Rik; all positively regulated), and 1 primed gene in PFC (*Twist1*, in opposite directions; Fig. 5l, n). Primed genes were largely not altered by ELS or adult stress alone, as indicated by minimal expression changes in the corresponding columns of each heatmap (Fig. 5c, e, g, i, k, m). Likewise, FC comparison between ELS + adult stress vs. Std-Ctl and vs. ELS-Ctl shows a majority of primed genes along the identity line, indicating that ELS alone did not alter primed genes (Fig. 5d, f, h, j, l, n). Magnitude of change was independent of base mean read count and significance of change (Fig. 5d, f, h, j, l, n). Some primed genes are annotated in Fig. 5c–h, and a full table of primed genes is included (Supplementary Data 3). The greatest number of primed genes were found in PFC for both male and female mice (Fig. 5g, h).

**ELS enriches for plasticity signatures.** We hypothesized that ELS might increase sensitivity to adult stress through increases in developmentally defined plasticity programs. Knockout of *Lynx1* in primary visual cortex preserves visual critical period plasticity beyond the juvenile critical period. Here, we use a previously computed transcriptional signature of *Lynx1* knockout associated with cortical plasticity, termed 'plasticity signature'[17,18]. We calculated a normalized enrichment score to compare stress and plasticity signatures in a threshold-free manner from the full DEG lists. Consistent with our hypothesis, transcriptional signatures of plasticity were significantly and positively enriched after ELS in male NAc (FDR = 0.0384, Fig. 6b), with trends in male VTA ($p = 0.0846$, Fig. 6a) and female PFC ($p = 0.0563$, Fig. 6c). The combination of early life and adult stress positively enriched plasticity scores selectively in female PFC (FDR = 0.0020). However, adult stress alone and the combination of stresses decreased plasticity enrichment in male PFC (FDR = 0.0277 and FDR < 0.0001, respectively), with a trend in male NAc ($p = 0.0364$).

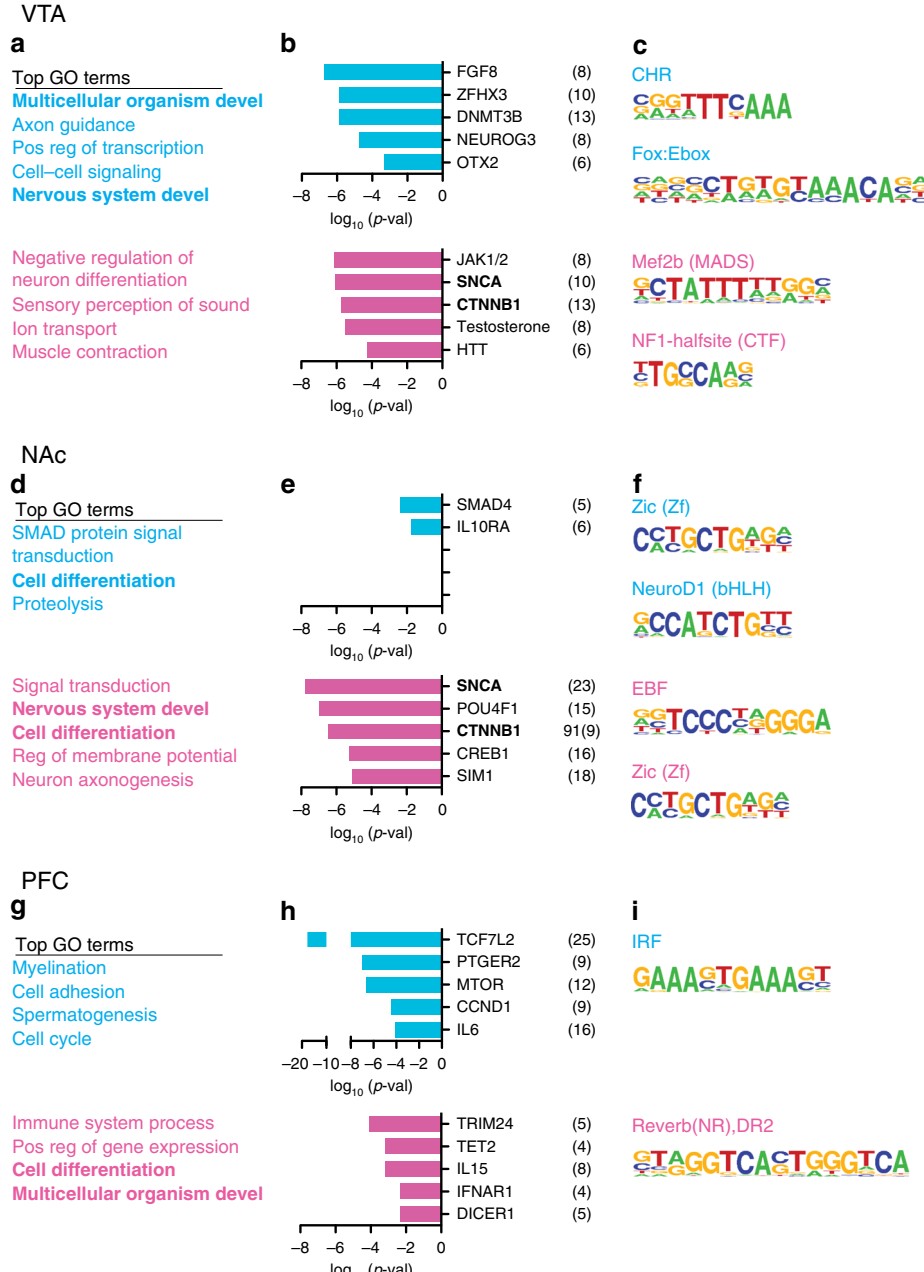

**Fig. 3** Predicted biological processes and transcriptional regulators altered by ELS. Top gene ontology (GO) terms enriched in ELS-control vs. standard-control DEGs from male (top, blue) and female (bottom, pink) VTA **a**, NAc **d** or PCF **g**. **b** Top predicted upstream regulators of ELS-control vs. standard-control DEGs from male (top, blue) and female (bottom, pink) VTA **b**, NAc **e** or PCF **h**. **c** Top predicted transcription factors and binding sequences in promotor regions of ELS-control vs. standard-control DEGs from male (top, blue) and female (bottom, pink) VTA **c**, NAc **f** or PCF **i**

## Discussion

Stress experienced early in life can change circuit development and response to later stress and reward, which are likely mediated in part by transcriptional changes in the brain early in life and sustained across the lifespan[7,19–22]. Here, we present systematic mapping of enduring ELS-induced transcriptomic changes in three reward-associated brain regions key to depression-like behaviors[23], in both male and female mice, and we further assess transcriptomic responses to a second stress in adulthood.

This work extends to female mice our earlier findings in male mice that ELS from P10–17 (or P10–20) induces susceptibility to subsequent adult stress. The mouse ELS paradigm used here was based on both rodent maternal separation studies and work demonstrating reduced nesting and/or bedding material induces

disordered maternal care among rodent dams[24,25]. We previously found that this same ELS procedure from P2 to 12 did not significantly alter depression-like behaviors in male mice, either before or after a second stress in adulthood, which may be due to immature stress-response circuitry or timing of molecular cascades that mediate adaptations to stressful stimuli[7,8,26]. In male mice, shifting the timing of ELS to P10–17 (or P10–20) also did not alter depression-like behaviors at baseline, but instead increased the risk that a second stress in adulthood would result in depression-like behaviors. These findings are in contrast to other rodent ELS paradigms that report baseline depression-like behaviors from early (<P14) postnatal stress in female mice[27], and evidence that ELS prior to P9 may in fact blunt hippocampal plasticity and dampen responses to additional stress[28,29]. Studies

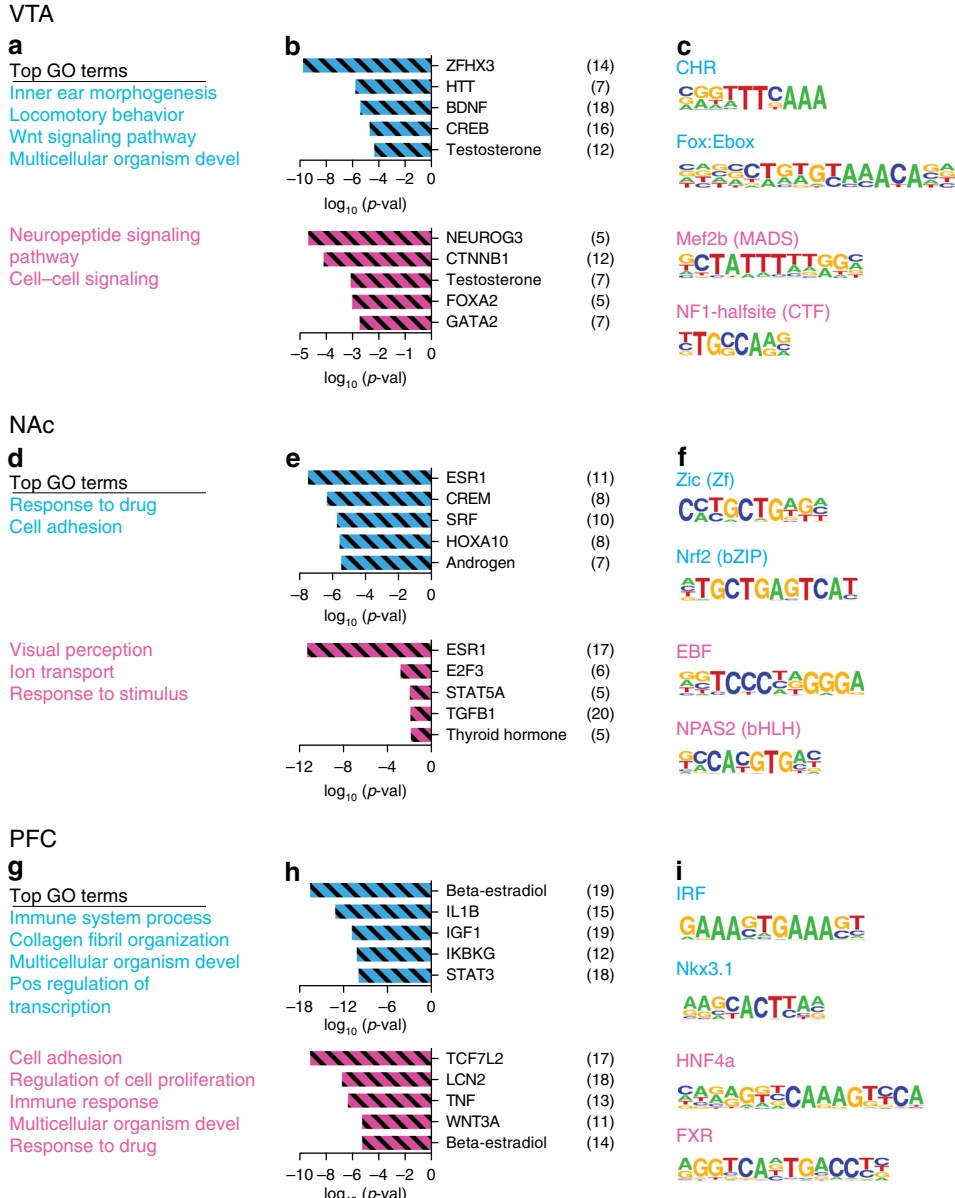

**Fig. 4** Predicted biological processes and transcriptional regulators altered by two hits of stress. Top gene ontology (GO) terms enriched in ELS + adult stress vs. ELS-control DEGs from male (top, blue) and female (bottom, pink) VTA **a**, NAc **d** or PCF **g**. **b** Top predicted upstream regulators of ELS + adult stress vs. ELS-control DEGs from male (top, blue) and female (bottom, pink) VTA **b**, NAc **e** or PCF **h**. **c** Top predicted transcription factors and binding sequences in promotor regions of ELS + adult stress vs. ELS-control DEGs from male (top, blue) and female (bottom, pink) VTA **c**, NAc **f** or PCF **i**. Adult stress was CSDS for males and STVS for females

of humans that experienced early life adversity have found earlier development of anxiety, depression, and other psychiatric illnesses, as well as latent vulnerability to these diseases[30–32]. For example, women abused as children are more likely to experience depression after additional adult abuse compared to women abused only as adults or only as children[5]. Key stress-responsive brain regions may mature faster in females than males[33], and ELS may alter stress-related behaviors earlier in females than males[34]. Thus, we tested the hypothesis that stress in the earlier rather than later postnatal period might have greater impact on depression-like behavior in female mice. However, the greatest differences in individual and composite behavioral measures were found after the combination of ELS from P10–17 with STVS (Supplementary Fig. 1a–c), consistent with our findings in male mice. Our current and previous findings of a late postnatal (>P10) stress sensitive period among both male and female mice are also

consistent with a lack of sex differences in the early postnatal period (<P9) of attachment sensitivity—coincident with a stress hyporesponsive period—which enables animals to attach to a caregiver to meet basic needs (feeding, thermoregulation, etc.) when animals are most vulnerable, regardless of stress associated with the caregiver[26].

The later postnatal ELS that we find increases vulnerability to adult stress may be due to a later sensitive period of catecholamine circuitry development[35] and later development of threat circuitry (see ref. [26]). Inhibition of noradrenergic cells from P10–21, but not P2–9 or P56–67, increased anxiety-like and depression-like behavior in adult mice[36]. Dopamine signaling in the striatum matures rapidly from P10 through adolescence and is required from P18 to 28, but not from birth or in adulthood, for normal physiological development and related behaviors[37,38]. Rodent amygdala begins to support fear-learning at P10, plasticity

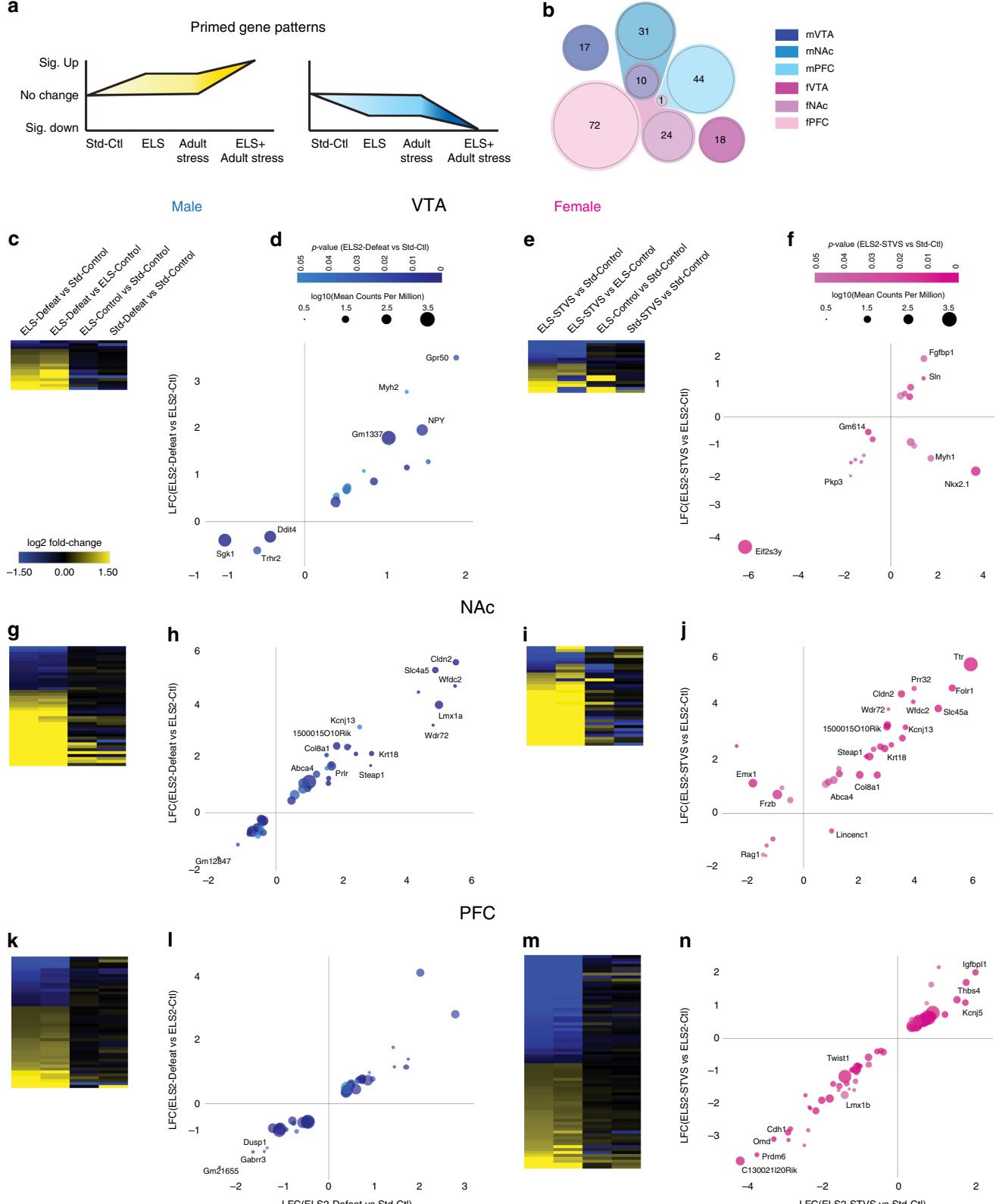

**Fig. 5** ELS primes a subset of genes in male and female brain reward regions. **a** Schematic depicting primed gene patterns. **b** Scaled multi-group Venn diagram of primed genes from each sex and brain region. Heatmaps depicting log$_2$-fold-change of primed genes in each comparison from male VTA **c**, female VTA **e**, male NAc **g**, female NAc **i**, male PFC **k**, and female PFC **m**. Primed genes plotted as log$_2$-fold-change of ELS + adult stress vs. standard-control by ELS + adult stress vs. ELS − control, colored by degree of significance and bubble size scaled by base mean expression level (log$_{10}$ of mean counts per million mapped reads) from male VTA **d**, female VTA **f**, male NAc **h**, female NAc **j**, male PFC **l**, and female PFC **n**

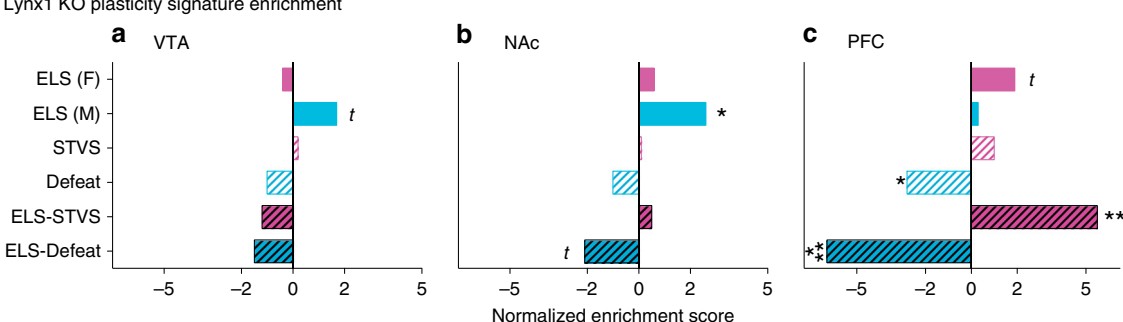

**Fig. 6** Enrichment of cortical plasticity signatures after stress. Normalized *Lynx1* KO plasticity signature enrichment score of indicated DEG lists from **a** VTA, **b** NAc, and **c** PFC. Each DEG comparison is to standard-control of the same sex and region. Significance of enrichment indicated by \*\*\*FDR < 0.001, \*\*FDR < 0.01, \*FDR < 0.05, *t* $p$ < 0.1

and long-term potentiation develop in rodent hippocampus in the second postnatal week, and connectivity between human amygdala and hippocampus appears to develop later in children[19,39,40]. The impact of ELS on these and other brain regions important for stress and threat-learning would be interesting to transcriptionally profile in future studies. While there are mixed reports on whether timing of ELS experience in humans influences later depression[41,42], it is also necessary to consider that mice are born premature relative to humans and for at least some brain systems rodent postnatal week-one is more similar to the second half of human gestation[43,44].

The latent behavioral stress sensitivity induced by ELS is useful for disentangling the enduring transcriptional impact of ELS associated with vulnerability, from neurobiological changes associated with depression-like behaviors per se or with acute stress. Across three brain reward-associated regions, we find broadly similar transcriptional changes after ELS and adult stress. These similarities were visualized using both thresholded (heatmap) and threshold-free (RRHO) analyses (Fig. 2). These findings corroborate and extend our previous findings of depression-like transcriptional patterns after ELS in the adult male VTA[7]. However, RNA-seq also revealed that adult stress after ELS does not simply amplify transcriptional differences observed after either stress alone, but instead results in unique transcriptional changes. Our results show that experience of stress in adulthood induces distinct transcriptional patterns depending on whether or not the mouse had previously experienced ELS, across multiple brain regions in both male and female mice. Similarly, chronic adolescent stress was found to induce distinct, latent patterns of gene expression in the hippocampus of male and female rats in response to an acute adult stress[45]. Together, these findings suggest that ELS alters transcriptional development of the brain-or, the lifelong potential (priming) of transcriptional regulation in response to subsequent stressful stimuli.

Given that male and female mice used in this study were generated in separate cohorts (see the "Methods" section), experienced different adult stresses (STVS or repeated social defeat), and tissue was processed separately, we have deliberately avoided direct comparisons of potential sex differences in response to stress as sex and cohort variables cannot be distinguished. RNA-seq in both human and mouse brain reveals substantial sex differences at baseline, as well as in human depression and stressed mice[13,46–48]. Nevertheless, it is still useful to consider here similarities in biological pathways affected by each stress or combination of stresses among male and female mice, even when distinct genes contribute to pathway enrichment. While the majority of predicted pathways and upstream regulators were unique, several GO terms related to brain and organism development were enriched in ELS DEG lists

across sexes and regions. These predictions suggest that ELS alters brain development across reward regions in both sexes, through distinct genes, but potentially common pathways related to neuronal outgrowth and synapse formation and signaling. Moreover, several regulators predicted in females have been previously implicated in depression-like behavior in male mice. Neuronal PAS-domain protein 2 (NPAS2) is a predicted transcriptional regulator after two hits of stress in female NAc (Fig. 4), and has been previously identified to mediate stress response through regulation of GABAA receptors in male NAc[49]. SNCA and CTNNB1 were both predicted upstream regulators of transcriptional response to ELS in female VTA and NAc (Fig. 3). SCNA (which encodes alpha-synuclein) regulates monoamine neurotransmitters and inhibits BDNF/TrkB signaling[50] (which is altered by early life and adult stress in mesocorticolimbic regions and associated with depression-like behavior[51,52]), and is also dysregulated in serum of depressed patients[53] and hippocampus of rats after stress across the lifespan[54,55]. *Ctnnb1* (which encodes ß-catenin) in male NAc D2 receptor subtype neurons was previously shown to mediate resilience to adult stress through DICER-mediated microRNA regulation[56]. DICER1 was also a top predicted upstream regulator in female PFC (Fig. 3h), suggesting common enduring molecular alterations across brain reward regions in response to stress. However, upregulation of these pathways in NAc was found to increase resilience in males, whereas these pathways are associated with stress susceptibility in females after ELS. These results therefore raise the possibility of prominent sex differences and/or timing of stress (developmental vs. adult) in the role played by CTNNB1 in stress responses.

Some similarities in transcriptional regulation between sexes were observed after two hits of stress compared to ELS alone. Most strikingly, steroid hormones and steroid hormone receptors are predicted upstream regulators after a second hit of stress in both males and females (Fig. 4). Of note, estrogen receptor-alpha (ESR1) is a predicted upstream regulator of transcriptional response to a second hit of stress in male and female NAc, but not to either stress alone. Increased frequency of rat maternal care (associated with lower anxiety-like behavior) increases *Esr1* expression and activity in the medial preoptic area of the hypothalamus in female offspring[57,58]. *Esr1* also drives a pro-resilient phenotype in NAc of male and female mice[59]. In contrast to this finding, in the current datasets ESR1 is predicted to be activated in female NAc after a second hit of stress, with no predicted direction in male NAc. This is similar to predicted upstream activation by ESR1 in hippocampus of rats exposed to the combination of chronic adolescent stress and acute adult stress[45]. It is possible that activation of ESR1 and its downstream gene targets in response to a second hit of stress represents a compensatory

mechanism to cope with additional stress, but additional time points are necessary to test this hypothesis.

Our previous work focused on enduring transcriptional changes after ELS that may underlie vulnerability to stress in adulthood, while the current analyses extends to latent transcriptional changes revealed by the second hit of stress, which we call primed genes (Fig. 5; Supplementary Data 3). Despite cohort differences, a portion of primed genes were common to both male and female NAc. Of the genes similarly primed up in male and female NAc, there was a significant gene ontology enrichment for ion transport ($p = 0.032$) which included *Kcnj13* (the voltage-gated inwardly rectifying potassium channel Kir7.1), *Steap1* (six-transmembrane epithelian antigen of prostate 1), and *Slc4a5* (electrogenic sodium bicarbonate cotransporter 4). Several of the primed genes identified by our analyses have been previously implicated in stress vulnerability and depression-like behavior in humans and rodents. *Sgk1* (serum- and glucocorticoid-inducible kinase 1) was primed down in male VTA. While *Sgk1* is increased in blood of depressed patients, it exerts a neuroprotective role under oxidative stress conditions which may be selectively suppressed in male VTA after ELS[60,61]. *Npy* has also been implicated in stress response (in different directions depending on region and acute vs. long-term sampling[62]) and is primed up in male VTA in our dataset and elevated in male and female hypothalamic regions after ELS from P2-9[63]. Only one primed gene, *Myrf* (myelin regulatory factor) in female PFC, has been associated with major depression in human GWAS studies[64]. Empirical testing is needed to determine whether manipulation of primed genes, alone or in combination, is sufficient to alter sensitivity to adult stress.

Lastly, we used these RNA-seq data to test the hypothesis that ELS enriches for established transcriptional signatures of cortical plasticity. In visual cortex, experience-dependent plasticity declines after a juvenile critical period concurrent with an increase in *Lynx1*, which binds nicotinic acetylcholine receptors[17]. *Lynx1* knockout preserves experience-dependent plasticity past the critical period into adulthood[17]. Our hypothesis was supported in male NAc, and to a lesser degree in male VTA and female PFC (Fig. 6). This suggests that one mechanism for increased sensitivity to future stress is through inappropriately preserved brain plasticity, in a sex-dependent and region-dependent manner. Enriched plasticity signatures were unique to stress experienced in the postnatal period, as adult stress alone either had no impact on plasticity or negatively impacted plasticity. However, after a second hit of stress, plasticity enrichment was reversed in male NAc and PFC, while it was augmented in female PFC. Thus, in males, a second stress exposure may be a trigger to close the extended plasticity period and decrease behavioral flexibility through decreased plasticity, leading to maladaptive behavior. In support of this theory, hippocampal synaptic plasticity in the form of stimulation-induced long-term potentiation (LTP) develops between P9 and P22, and a single day of maternal deprivation increases plasticity in juvenile males, which can then be suppressed by corticosterone in adult males[65]. Stress in adulthood similarly enhances plasticity in VTA and NAc, though the long-term impact of developmental stress on physiological plasticity in VTA and NAc remain to be tested[66,67]. Both ELS and adult stress impair plasticity in male PFC, consistent with our plasticity signature findings[68,69].

The two-hit stress paradigm utilized in female and male mice in the present study is useful for studying neurodevelopmental adaptations to ELS associated with later vulnerability to adult stress and the development of depression-related behavioral abnormalities. In particular, these comprehensive RNA-seq datasets provide a foundation for future research to test the causal role of specific molecular pathways implicated in ELS-induced stress sensitivity and priming in both sexes.

## Methods

**Animals**. C57BL/6J mice were maintained on a 12-h light/dark cycle (lights on at 7 a.m.) with ad libitum access to food and water. All experiments were conducted in accordance with the guidelines of the Institutional Animal Care and Use Committee at Mount Sinai and of the Society for Neuroscience. All behavioral testing occurred during the animals' light cycle. Experimenters were blind to experimental group, and order of testing was counterbalanced during behavioral experiments.

All mating occurred in-house to prevent shipping stress during gestation. Two nulliparous C57BL/6J female mice (Jackson) were mated with one male in our animal facility. Males were removed after 5 days and females were separated into individual cages 1–3 days prior to parturition. Litters were weighed and counted and cages cleaned on the day of birth (PND0) but otherwise undisturbed. Cages were cleaned with minimal disruption to the litter once/week. Offspring were weaned at postnatal PND21 with males and females weaned separately into cages of 3–5 mice, keeping littermates together and only combining pups from different litters and of the same experimental condition to maintain three or more mice/cage. Male and female mice included here were from separate cohorts.

**Stress paradigms**. ELS consisted of a combination of maternal separation and limited nesting[25,70,71] from P10–17 (females; Fig. 1a) or from P10–20 (males; Fig. 1l)[7]. Litters were randomly assigned to standard-rearing (Std) or ELS groups. Pups were separated together as a litter to clean cages with distinct bedding for 3–4 h/day at random times each day during the light cycle to minimize predictability and habituation. EnviroDri nesting material was depleted to 1/3 of standard-reared cages during the days of separations and then restored on the final day. Early-ELS was identical but from P2 to 10, and separation cages were placed on warming pads to maintain constant low temperature (32–34 °C), since pups at that age cannot thermoregulate on their own.

Adult mice were assigned randomly to control or adult stress conditions, and littermates were assigned to different groups for within-litter controls. Adult stress for male mice consisted of chronic social defeat stress (CSDS), as previously reported[7,12]. Adult stress for female mice consisted of 3 days of STVS[13] (Fig. 1a) in order to test whether ELS increased vulnerability to stress that would not normally result in depression-like behavior for standard-reared mice. We used STVS in female mice because, at the time of this study, a social defeat procedure had not yet been validated for female C57BL/6J mice. On 3 consecutive days, female mice experienced: 100 random mild foot shocks at 0.45 mA for 1 h (10 mice to a chamber), a tail suspension stress for 1 h, and restraint stress in a 50 mL conical tube for 1 h. Mice were individually housed following the final stressor, and behavioral testing began the next day.

**Behavioral testing**. Behavior of male mice was characterized and reported previously[7]. Female depression-like behaviors were tested in one cohort used for RNA-seq [($n = 13$ (Std-Ctl), 15 (Std-STVS), 12 (ELS-Ctl), 13 (ELS-STVS)] and in an additional replication cohort which included early-ELS exposed mice [$n = 10$ (Std-Ctl), 11 (Std-STVS), 9 (ELS-Ctl), 9 (ELS-STVS), 11 (early ELS-Ctl), 11 (early ELS-STVS)]. Beginning the day after the last day of adult stress, one behavioral test was conducted per day for 3 consecutive days, in the order described below, with the exception of sucrose preference which spanned behavioral testing days.

Splash test was performed consistent with previous studies[13,72]. The test was performed under red light (230 V, 15 W). Mice were placed in an empty cage and sprayed on the back with a 10% sucrose solution two times. Behavior was video recorded for 5 min The latency to begin grooming the back, and the total amount of time grooming thereafter (back, hands, and face) was scored by hand by an observer blind to experimental group. Increased latency and decreased total grooming time were considered indicators of depression-like or anxiety-like behavior[13].

Sucrose preference, a measure of anhedonia-like behavior in mice, was assessed in a two-bottle choice test[73]. Upon single-housing, mice were acclimated overnight with two bottles of drinking water (50 mL conical tubes fitted with spouted rubber tops). The next day after conclusion of the splash test, water in one bottle was replaced with a 1% sucrose solution and both bottles weighed. Bottles were weighed again daily at the beginning of the light cycle for 2 days. Bottle locations were switched at each measurement to prevent location habituation. Percent sucrose preference was calculated as amount (g) sucrose solution consumed over total amount (g) of water and sucrose consumed.

Novelty suppressed feeding testing was adapted from previous work[10]. Mice were food restricted overnight before testing. On the day of testing, mice habituated to the testing room for at least 1 h. Under red light conditions, mice were then placed into a $40 \times 40 \times 20$ cm arena with wood-chip bedding covering the floor and a single standard chow food pellet in the center of the arena. Mice were placed in the corner of the box, and the latency to eat was hand-scored for up to 5 min (first cohort) or 10 min (second cohort). A video-tracking system (Ethovision, Noldus) measured locomotor activity. At the end of the test, mice were transferred to their home cage in standard lighting conditions, and the latency to eat was recorded by hand. A ratio of latency to eat in the novel arena/home cage was then calculated.

For the forced swim test, mice were individually placed in beakers of $25 \pm 1$ °C water for 6 min with ambient lighting. Immobility was assessed by a video-tracking system as a measure of depression-like behavior (Ethovision, Noldus).

A composite behavior outcome score was calculated based on all tests of depression-like and anxiety-like behavior, as previously calculated for male

behavior[7]. For the initial behavioral cohort (Fig. 1j and Supplementary Fig. 1), the behaviors included ratio of time to eat in novel vs. home cage environments (>mean + 1SD of standard-control scored as depression-like), latency to immobility in the forced swim test (<mean − 1SD), sucrose preference (<50%), and splash test grooming duration (< mean − 1SD). Forced swim was not included in the replication cohort. A percent of tests on which each mouse met depression/anxiety-like criteria was then calculated. This analysis was done in a post-hoc and exploratory manner, but was part of the initial plan for the study based on our published findings[7].

**Estrous cycle.** Vaginal swabs were taken from all female mice starting one day prior to behavioral testing and on the day of euthanasia. 15 μL of sterile PBS was gently pipet in and out of the vagina and smeared on a glass slide. Estrous state was immediately determined with a light microscope by cytology of nucleated, cornified, or leukocytic cells, taking into account the previous day's cytology. Because we did not attempt to artificially synchronize cycle state across females, there were uneven distributions of cycle states across groups on any given test day. Notably, however, nearly all mice entered diestrus following forced swim testing and remained in diestrus on the day of euthanasia. As such, tissue samples for RNA-seq are primarily from the diestrus stage.

**Statistical analysis of behavior.** All animals from a litter experienced the same early life conditions. Siblings were randomly assigned to different adult conditions. Subject number occasionally varied within a group between outcome measures due to improper video recording or leaked sucrose preference bottles. Outliers, defined by values more than two standard deviations from group mean, were excluded, which represented <3% of all observations. Prism (version 8, GraphPad) and SPSS (IBM, v25) were used for all graphing and statistical analysis of behavior. Significance thresholds were set at $p < 0.05$. Main effects and interactions were analyzed by two-way ANOVA. Two-tailed Student's t-test was used for comparison between individual groups if a main effect or interaction was found. The impact of estrous cycle state on behavior was assessed by using cycle state taken on the day of each behavioral test as a factor in ANOVA analyses. There were no main effects of cycle state for any behavior.

**RNA extraction and RNA-sequencing library preparation.** Samples from male brain were from mice whose behavior was reported previously (n = 4–6 per group)[1]. Samples from female brain were from mice whose behavior is reported in the initial cohort here (Fig. 1; n = 5–6 per group).

Adult mice were cervically dislocated directly from the home cage 1 day after the final behavioral test. Brains were removed rapidly, placed into ice-cold PBS, and sliced into 1 mm-thick coronal sections in a slice matrix. Bilateral punches were made from VTA (16 gauge), NAc (14 gauge), and PFC (12 gauge) and flash-frozen in tubes on dry ice. Total RNA was isolated with TriZol reagent (Invitrogen) and purified with RNeasy Micro Kits (Qiagen). All RNA samples were determined to have A260/280 values ≥ 1.8 (Nanodrop); samples for RNA-seq had RIN values > 9 (BioAnalyzer, Agilent). 500 ng of purified RNA was used to prepare libraries for sequencing using the Truseq mRNA library prep kit (Illumina RS-122-2001/2). As previously reported, male VTA and NAc samples were pooled 3 animals/sample (pooling based on similar behavioral measures) prior to library preparation, and were sequenced on an Illumina Hi-seq machine with 50-nt single-end reads in the Mount Sinai Genomics Core Facilities[7]. Male PFC and female VTA, NAc, and PFC samples were prepared from individual animals (4–6 independent samples/group), and sequenced with 125-nt single-end reads at Beckman Coulter Genomics (currently Genewiz). Samples were multiplexed to produce >30M reads/sample. All RNA-seq files are available through Gene Expression Omnibus (GEO), accession GSE89692.

**qPCR.** RNA was extracted as above from VTA, NAc, or PFC tissue punches from independent cohorts of ELS and Std male and female mice (n = 5–8 per group) and converted to cDNA using either SuperScript III (Invitrogen 18-080-400) or High-Capacity cDNA Reverse Transcription kits (Applied Biosystems 43-688-14). Primer efficiency and specificity were verified; sequences are given in Supplementary Data 4. Real-time semi-quantitative qPCR was performed using SybrGreen Fast master mix and standard cycling conditions on a QuantStudio 5 and analyzed by the $2^{-\Delta\Delta Ct}$ method with Hprt as a control gene. Genes were semi-randomly chosen across levels of significance, brain regions, and sexes. Only protein-coding genes with high-quality primer validation were included.

**RNA-seq differential expression analysis.** RNA-seq reads were aligned to mouse genome NCBI37 (mm9) using Tophat2[74]. The average mapping rate was 93% (Supplementary Data 1). Uniquely aligned short reads were counted using HTSeq-counts. Principal component analysis was used to detect outliers for removal, although none were identified. Normalization and differential analysis was performed using DESeq2[75].

All genes included in differential gene expression analysis had a base mean expression >2. Significance was set at uncorrected $p < 0.05$ for broad pattern identification (Fig. 2). A FC threshold was set at >30% ($\log_2$ FC, LFC > |0.3875|) for each comparison. DEGs for each comparison represented in Venn Diagrams (Fig. 2) are limited to protein-coding genes and are reported in Supplementary

Data 2. Gene lists for heatmaps included a union of all DEGs from any one comparison with matched LFC values regardless of significance in the other comparisons. Heatmaps were generated using Morpheus (Morpheus, https://software.broadinstitute.org/morpheus) and clustered by one minus Pearson correlation and average linkage. Cell-type enrichment analysis used curated gene lists in which a gene was considered a cell-type-specific marker if average expression met the stringent criteria of >10× the average background expression levels of the remaining cell types in that region[16]. Enrichment of DEGs with these cell-type marker lists was determined as an odds ratio of list overlap using the GeneOverlap tool in R[76]. Functional annotation for gene ontology of biological processes was performed using DAVID Bioinformatics Resource 6.7, with ≥5 genes and ease = 0.05, reporting only non-redundant categories[77,78]. Upstream regulator predictions were made using the May 2018 release of ingenuity pathway analysis (IPA, Qiagen). Predicted upstream regulators included top predicted genes, cytokines, or endogenous chemicals, made from ≥3 up-regulated or down-regulated DEGs. HOMER motif analysis was used to predict differential transcriptional regulators including transcription factor-binding sites and transcriptional elements from enriched 8–10 base sequences located within −2000 to +1000 bases of the TSS of DEGs[79].

Primed genes (Supplementary Data 3) were defined as genes in the ELS + adult stress group that were significantly ($p < 0.05$) different from standard-control expression levels with LFC > |0.3875|, and also significantly ($p < 0.05$) different from both ELS-alone and adult stress-alone groups, in either direction without FC cutoff (Fig. 5a). Overlap among the six lists were plotted using the webtool InteractiVenn.net[80]. LFC expression differences were plotted with Plotly[81] in R with color density indicating significance level and bubble size representing base expression level ($\log_{10}$ mean counts per million).

**Rank-rank hypergeometric overlap (RRHO) RNA-seq analysis.** Full threshold-free differential expression lists were ranked by the $-\log10(p\text{-value})$ multiplied by the sign of the fold change from the DESeq2 analysis, filtered to all genes with a mean base expression of >2 RPKM to avoid low-expression artifacts. RRHO was used to evaluate the overlap of differential expression lists between stress comparisons[82–84]. A two-sided version of this analysis was used to test for coincident and opposite enrichment[15]. RRHO difference maps were produced for pairs of RRHO maps (ELS-Control vs. Standard-Control compared to adult stress-Control vs. Standard-Control, and adult stress-Control vs. Standard-Control compared to ELS + adult stress vs. ELS-Control) by calculating for each pixel the normal approximation of difference in log odds ratio and standard error of overlap between each matched comparison. This Z score was then converted to a P-value and corrected for multiple comparisons across pixels.

Male and female tissue sequenced in this study were from separate cohorts, making a direct comparison of male vs. female samples invalid as we cannot separate sex effects from cohort effects. Having said that, we observed 3–15% overlap of DEG's between males and females in each of the three brain regions studied for each comparison, which is consistent with very small overlap seen between depressed human males and females, as well as between male and female mice subjected to the same stress procedure and analyzed within the same cohort[13,46,47].

**RNA-seq plasticity signature analysis.** $Lynx1^{-/-}$ plasticity signatures were generated from publicly available RNA-seq data derived from adult (>P60) primary visual cortex tissue punch of male $Lynx1^{-/-}$ vs. WT mice (GEO: GSE89757[18]). Briefly, we used Limma[85] to quantile normalized raw microrray probe-level data and RankProd[86] to compute rank-based differential expression of mouse genes retaining genes with a false discovery rate < 0.05[87], which we mapped to Entrez IDs to yield a 114 gene $Lynx1^{-/-}$ transcriptional signature.

Genes differentially expressed across the entire transcriptome of early life and adult stress signatures were converted to Entrez IDs and unmapped and unexpressed genes were removed. A molecular matching score was calculated between the plasticity signature and a given stress signature by summing the logFC expression values in a stress signature that are shared with genes decreased in the plasticity signature and subtracting them from the sum of logFC expression values shared with genes increased in the plasticity signature to yield a summary measure of concordance between stress and plasticity signatures. High molecular match scores (>0) indicate a given stress signature mimics the plasticity signature whereas low molecular match scores (<0) indicate a given stress signature induces gene expression that opposes the plasticity signature.

To estimate the P value of molecular match scores (M) between a given stress and plasticity signature, we calculated an empirical P value for M given n = 10,000 permutations of M ($M_{perm}$), computed by shuffling the gene labels of the stress signature and recalculating M. The generalized Pareto distribution[88] was used to improve accuracy of the P value estimation and multiple hypothesis tests were adjusted for using the false discovery rate[87]. To compare M for a given plasticity signature across the stress signatures, we normalized M with $M_{perm}$ according to

$$\frac{M - \bar{M}_{perm}}{\sqrt{\sum_{i=1}^{n} \frac{(M_{perm_i} - \bar{M}_{perm})^2}{n-1}}}$$

to yield a normalized score M (the effect size). M is computed similar to the approach by Zhang and Gant[89]. Analyses were completed in the R programming language (v 3.2.2).

**Reporting summary**. Further information on research design is available in the Nature Research Reporting Summary linked to this article.

## Data availability

All RNA-seq data is publicly available through the Gene Expression Omnibus, Accession number GSE89692 [https://www.ncbi.nlm.nih.gov/geo/query/acc.cgi?acc=GSE89692]. The source data underlying Fig. 1 and Supplementary Figs. 1 and 2 are provided as a Source Data file.

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

## Acknowledgements

We would like to thank Daniel A. Porada, Gabriella Fricklas, and Delaney Dalldorf for their work scoring behavior. This work was supported by funding from the National Institutes of Health (K99 MH115096 to C.J.P.; P50 MH096890 and R01 MH051399 to E.J.N.) and the Hope for Depression Research Foundation.

## Author contributions

C.J.P. and E.J.N. designed the studies. C.J.P., M.S., R.C.B., H.G.K., H.M.C., B.P., and A.B.C. performed the experiments. C.J.P., M.S., A.R., and I.P. performed the data analysis with input from L.S., H.M., and J.D. C.J.P., M.S., and E.J.N. took part in interpretation of the results. C.J.P. wrote the manuscript. All authors approved the manuscript.

## Competing interests

The authors declare no competing interests.
