## [Peer Review File · Nature Communications]

Reviewers' comments:

Reviewer #1 (Remarks to the Author):

The authors used RNA-sequencing in the ventral tegmental area, nucleus accumbens, and prefrontal cortex in male and female mice to test the hypothesis that adult stress is distinctly represented in the brain's transcriptome, depending on a history of early life stress (ELS). The results showed that biological pathways that were disrupted after ELS were associated with greater behavioral sensitivity to stress, that there are putative transcriptional regulators of the effect of ELS on the adult stress response, and that subsets of primed genes are specifically associated with latent depression-like behavior. They also provided transcriptomic evidence that ELS increases the sensitivity to future stress by enhancing known programs of cortical plasticity. This is a very important study that shows similar transcriptional changes after ELS and adult stress across three brain reward-associated regions. Particularly innovative is the use of a two-hit stress paradigm that has significant translational validity with the human condition. Particularly interesting are the RNA-seq data that showed that adult stress after ELS did not simply amplify transcriptional differences that were observed after either stress alone but resulted in unique transcriptional changes. This reviewer has only minor issues for the authors to address.

1. It is not entirely clear whether the female-only replication was significant for the overall depression score. Could the authors explain this better in the Results section?
2. The discussion does broach the subject of other brain areas that are often associated with overactivation in stress and posttraumatic stress syndrome, namely the amygdala. It is regrettable that that this structure was not included in the present study. I am assuming the stress system will be a separate study (?). The authors may wish to elaborate a bit more on the potential role of the amygdala in this plasticity.
3. Despite the fact that the authors are probably correct, it is quite dangerous to claim to be the "first" to do anything in science, so I recommend removing those claims from the manuscript.

Reviewer #2 (Remarks to the Author):

In the manuscript, "Early life stress alters transcriptomic patterning across reward circuitry in male and female mice", the authors investigate the effects of early life stress, with or without an adult stress exposure, on: 1) anxiety-/depressive-like behavior in female mice; 2) large-scale gene expression in 3 mood/reward-related brain regions in both male and female mice; and 3) use biostatistical/bioinformatics approaches to further probe the RNA-seq data. The authors report that early life stress, in conjunction with an adult stress, increases anxiety-/depressive-like behavior in female mice; this result is consistent with what these authors previously reported for male mice. Next, the authors report which genes are affected by ELS +/- adult stress. Finally, the authors perform pathway, upstream regulator, transcription factor, and threshold-free analysis to further probe the RNA-seq data. Since early life stress increases the risk for adult mood disorders, investigation of the mechanisms that might prime the system to be more susceptible to adult stress are highly relevant; since mood disorders are more common in females, it is great that the authors performed their studies in both sexes. However, the main concern I have with this paper is that it is largely descriptive, without validation of the key findings.

Major Comments:

- 1) The major issue I have is that the authors do not functionally manipulate a pathway identified via RNA-seq and determine effects on mood-related behavior. This evidence is necessary to strengthen the main conclusion.
- 2) The authors have generated an extensive RNA-seq dataset, but have not validated top hits,

which I believe is essential (e.g., by qPCR).

3) In most cases, the authors report 2-group comparisons in the behavioral data only if there is a significant interaction. However, there are a couple of places in which 2-group t-tests were performed without a significant interaction. For instance, this is the case in Figure 1f and 1j. These should be indicated as exploratory since the interaction was not significant.

4) The calculation for "composite depression score" seemed arbitrary to me. Is this a method used by others? If so, please provide a citation.

5) Based on the cell type-specific predictions, the authors conclude that there is little cell type-specificity. However, to me, there does seem to be some consistent findings. For example, "BrainstemCholin" comes up as a hit for both male and female VTA, "Microglia" comes up as a hit for male VTA and NAc, and "Oligo" comes up as a hit for PFC in both sexes. I suggest pointing out these consistencies and including discussion of these findings in the discussion.

6) I suggest having the same scale for all RRHO plots in Figure 2. I think that this might end up changing the authors interpretation of the RRHO results. I think that the strong signal in some plots (e.g., in Figure 2w (left)) might drown out signal in other plots. For instance, if Figure 2w (right) had the same scale as 2o (right), the opposite effect in females might pop up; the same direction effect might also pop up in male PFC (Figure 2t) if the scale wasn't so high.

7) For the GO pathway analyses in Figure 3, the authors conclude that there is overlap in pathways affected in males and females. However, to me, there seems to be only a small amount of overlap. Instead of concluding that stress results in different sets of affected genes, but the same pathways, I believe that the result is actually more mixed than the authors conclude. How might this affect the conclusions of the manuscript?

8) In a couple of places, I had to read the results several times to understand the findings/interpretations. This was especially true for the second set of RRHO plots (Figure 2d,g,l,o,t,w (right plots)) and for the comparisons made to determine "primed genes". I suggest adjusting the text to make it easier to digest (in results and in discussion).

9) Does RRHO yield similar results as those reported in Figure 6?

10) The authors state in the methods that they determined phase of estrous cycle in females, but do not include this information in any analyses. Does phase of cycle influence behavior and/or gene expression results? I think this is an especially important point since the authors identify gonadal hormones and receptors and the top upstream regulators in Figure 4.

11) I am confused about the methods for RNA-seq (but maybe I missed something). The authors state first that male VTA and NAc samples were pooled 3 animals/sample, but then say that male and female VTA, NAc, and PFC samples were prepared from individual animals (4-6 independent samples/group).

Minor Comments:

1) While I believe that the behavioral replication cohort is important, the text associated with it could probably be majorly reduced. This text could instead be placed in the figure legend for Supplementary Figure S1.

2) The text for Figure 5 axes is too small to read.

3) The heatmaps in Figure 2 seem kind of washed out.

Reviewer #3 (Remarks to the Author):

In this paper, the authors extended their previous finding of the effect of early life stress on the response to social defeat stress in adult male mice, and associated transcriptional changes in ventral tegmental area (VTA). In this study, the authors performed similar experiments in female mice, and the effect of earlier stress was also examined in female mice. They also examined two brain areas, nucleus accumbens and prefrontal cortex, in addition to VTA. First, they performed behavioral analyses to show that there is an interaction between early life stress and stress in adults. Next, they performed RNAseq analyses and found that early life stress affected the stress-induced transcriptome changes in adults. They performed a number of bioinformatics analyses to interpret the obtained data.

Though the obtained results are robust and potentially interesting, the interpretation is sometimes arbitrary. No additional experiments to support the causal relationship between behavioral alterations and transcriptome changes were done, and thus this is a descriptive study. However, the experiments are well thought out, and the data are precious. The authors have made the data publically available, which will be useful for further studies.

The paper should be published after considering several minor issues listed below.

- 1) Abstract. "associated with latent depression-like behavior". This is misleading because the interaction between early life stress and defeat was seen only for anxiety scores and the composite score of depression- and anxiety- related behavioral tests.
- 2) Introduction. The authors state that early life stress (ELS) increases the risk of depression by two- to four- fold. This is overestimated. Retrospective assessment is affected by current mental status, and a prospective study reported lower odds ratio (Widom et al, Arch Gen Psychiatry 2007).
- 3) Results. The authors generated "a composite depression score". However, this includes the novelty-suppressed feeding test, which is a test for anxiety. Thus, the score should be named as "a composite depression and anxiety score".
- 4) It seems that the use of a composite score was not intended at the initiation of this study, but it was used because there was no significant interaction for the results of the tests for depression-like behavior. If so, the authors should state that this analysis was done in the post hoc manner and has exploratory nature.
- 5) P4. The last sentence of 1st paragraph. The authors use the term. "transcriptional priming." This term is misleading. Observed difference of transcriptome may be caused by the structural differences such as neural morphology, cellular composition, etc. The term "transcriptional priming" gives impression that the observed transcriptome difference is reflecting some intracellular molecular biological phenomena.
- 6) P7. Left column, Line 10. "plasticity signature". This is too general, and misleading. Actually, it is the "transcriptome changes associated with Lynx1 knockout".

Response to Reviewers NCOMMS-19-13687

Dear Reviewers,

Thank you for your very thoughtful reviews and the opportunity to revise our manuscript. We believe that the manuscript is much improved with the suggested edits and additional data. We have included the main critiques from reviewers here (black font) and responded to each comment (blue font). Additional/edited text within the revised manuscript itself is indicated in red font.

Thank you very much for your continued consideration of this manuscript.

Catherine Peña and Eric Nestler (on behalf of all the authors)

Reviewer #1 (Remarks to the Author):

This is a very important study that shows similar transcriptional changes after ELS and adult stress across three brain reward-associated regions. Particularly innovative is the use of a two-hit stress paradigm that has significant translational validity with the human condition. Particularly interesting are the RNA-seq data that showed that adult stress after ELS did not simply amplify transcriptional differences that were observed after either stress alone but resulted in unique transcriptional changes. This reviewer has only minor issues for the authors to address.

1. It is not entirely clear whether the female-only replication was significant for the overall depression score. Could the authors explain this better in the Results section?

The “composite behavior outcome score” (renamed as suggested by Reviewer 3) was not significant in the replication cohort, as altered NSF behavior was the most robustly replicated behavior among females. We have added a sentence to the results to clarify this. As suggested by Reviewer 2, we have also simplified the main text associated with the replication cohort and moved the statistical analysis to the Supplemental Figure 1 legend.

2. The discussion does broach the subject of other brain areas that are often associated with overactivation in stress and posttraumatic stress syndrome, namely the amygdala. It is regrettable that that this structure was not included in the present study. I am assuming the stress system will be a separate study (?). The authors may wish to elaborate a bit more on the potential role of the amygdala in this plasticity.

Regrettably, we could not profile all brain regions that have been linked to stress, fear, or fear learning, including the amygdala, hippocampus, and hypothalamus, among others, although it would certainly be interesting and valuable to assay these structures in future studies. We have added a sentence to the Discussion towards this point.

3. Despite the fact that the authors are probably correct, it is quite dangerous to claim to be the “first” to do anything in science, so I recommend removing those claims from the manuscript.

Thank you - we have removed “first” from the text.

Reviewer #2 (Remarks to the Author):

In the manuscript, “Early life stress alters transcriptomic patterning across reward circuitry in male and female mice”, the authors investigate the effects of early life stress, with or without an adult stress exposure, on: 1) anxiety-/depressive-like behavior in female mice; 2) large-scale gene expression in 3 mood/reward-related brain regions in both male and female mice; and 3) use biostatistical/bioinformatics approaches to further probe the RNA-seq data. The authors report that

early life stress, in conjunction with an adult stress, increases anxiety-/depressive-like behavior in female mice; this result is consistent with what these authors previously reported for male mice. Next, the authors report which genes are affected by ELS +/- adult stress. Finally, the authors perform pathway, upstream regulator, transcription factor, and threshold-free analysis to further probe the RNA-seq data. Since early life stress increases the risk for adult mood disorders, investigation of the mechanisms that might prime the system to be more susceptible to adult stress are highly relevant; since mood disorders are more common in females, it is great that the authors performed their studies in both sexes. However, the main concern I have with this paper is that it is largely descriptive, without validation of the key findings.

Major Comments:

1) The major issue I have is that the authors do not functionally manipulate a pathway identified via RNA-seq and determine effects on mood-related behavior. This evidence is necessary to strengthen the main conclusion.

We agree that causal manipulation is the gold standard to test validity of sequencing predictions. As such, we tested one prediction from one comparison from one brain region (the prediction that OTX2 was an upstream regulator of ELS vs Std transcriptional changes in male VTA) in a previous manuscript (Peña et al., *Science* 2017). However, here we focused on presenting broad patterns across multiple brain regions and the two sexes. We believe that causal testing of predictions from each comparison/region/sex would unnecessarily delay sharing of this rich dataset, and further complicate the broad patterns that we are able to deduce here.

2) The authors have generated an extensive RNA-seq dataset, but have not validated top hits, which I believe is essential (e.g., by qPCR).

We agree and, as suggested, performed qPCR for a subset of genes from the adult ELS vs Std comparisons for each sex and brain region. We were unable to use RNA originally used to generate the sequencing libraries themselves (as done for validation by many RNAseq studies), and instead attempted replication from samples from entirely independent cohorts of mice—a much higher bar. While not all differences reached statistical significance, 72% of the 18 genes analyzed were altered in the same direction as predicted by the sequencing data. These new findings suggest that the broad patterns of transcriptional change following ELS hold across cohorts. We have added these data to the Methods, Results, and Supplemental Figure S2, with PCR primers used given in Supplemental Table 4.

3) In most cases, the authors report 2-group comparisons in the behavioral data only if there is a significant interaction. However, there are a couple of places in which 2-group t-tests were performed without a significant interaction. For instance, this is the case in Figure 1f and 1j. These should be indicated as exploratory since the interaction was not significant.

We have clarified that these t-tests were exploratory given only main effects and not interactions. This text is added to the Figure 1 legend along with the associated statistics.

4) The calculation for “composite depression score” seemed arbitrary to me. Is this a method used by others? If so, please provide a citation.

The reviewer’s point is well taken. We previously generated a “composite behavior outcome score” (renamed as suggested by Reviewer 3) in an earlier paper including male behavior only (Peña et al., 2017) and included the measure in this manuscript to be consistent in the reporting of behavior for both sexes. However, we acknowledge it is not widely used. To our knowledge, there is not a better metric for evaluating behavior across a battery of multiple behavioral tests, which was the original goal of the metric.

5) Based on the cell type-specific predictions, the authors conclude that there is little cell type-specificity. However, to me, there does seem to be some consistent findings. For example,

“BrainstemCholin” comes up as a hit for both male and female VTA, “Microglia” comes up as a hit for male VTA and NAc, and “Oligo” comes up as a hit for PFC in both sexes. I suggest pointing out these consistencies and including discussion of these findings in the discussion.

We have added these similarities to the text. We did not initially want to place too much emphasis on these similarities, as they were inconsistent as to which stress comparison showed enrichment, but it is important to point out.

6) I suggest having the same scale for all RRHO plots in Figure 2. I think that this might end up changing the authors interpretation of the RRHO results. I think that the strong signal in some plots (e.g., in Figure 2w (left)) might drown out signal in other plots. For instance, if Figure 2w (right) had the same scale as 2o (right), the opposite effect in females might pop up; the same direction effect might also pop up in male PFC (Figure 2t) if the scale wasn't so high.

We have taken the Reviewer's suggestion to normalize scales across brain regions and not just across sexes. The main conclusions remain for VTA and NAc, but we have updated two points in the text: 1) there is clearer opposite regulation in female PFC by adult stress (with vs without prior ELS), and 2) there is similar regulation in male PFC by adult stress.

7) For the GO pathway analyses in Figure 3, the authors conclude that there is overlap in pathways affected in males and females. However, to me, there seems to be only a small amount of overlap. Instead of concluding that stress results in different sets of affected genes, but the same pathways, I believe that the result is actually more mixed than the authors conclude. How might this affect the conclusions of the manuscript?

We agree with the reviewer's interpretation, and our main conclusion from these analyses is that **“ELS induces transcriptional alterations via distinct regulatory pathways in males and females”** as stated in the Results heading. Nevertheless, we thought it relevant to point out the few similarities that were present, as with the cell-type enrichment analysis suggested by the reviewer. We have added text to the Discussion to reiterate the overall uniqueness, while still discussing relevance of the few common pathways in the context of previous studies of stress and depression.

8) In a couple of places, I had to read the results several times to understand the findings/interpretations. This was especially true for the second set of RRHO plots (Figure 2d,g,i,o,t,w (right plots)) and for the comparisons made to determine “primed genes”. I suggest adjusting the text to make it easier to digest (in results and in discussion).

Thank you for the suggestion to make the text clearer. We have added a few clarifying sentences, including: “In other words, the impact of adult stress was transcriptionally different depending on the history of ELS.” We hope that additional simplification and clarification in these sections has made the findings and interpretations clearer.

9) Does RRHO yield similar results as those reported in Figure 6?

The strength of the RRHO analysis is that it is threshold-free across all expressed genes, while the plasticity signature enrichment analysis necessarily uses a curated (“thresholded”) list of “plasticity” genes, on the order of dozens of genes. Unfortunately, this mismatch means the RRHO visualization is not valid for this particular comparison.

10) The authors state in the methods that they determined phase of estrous cycle in females, but do not include this information in any analyses. Does phase of cycle influence behavior and/or gene expression results? I think this is an especially important point since the authors identify gonadal hormones and receptors and the top upstream regulators in Figure 4.

This is an important point. We tested the hypothesis that high estradiol (estrus state) would fail to produce anxiolytic/antidepressive-like effects among ELS mice, as per recent evidence from the Kundakovic lab after adolescent stress (Jaric et al., *Frontiers in Mol Neurosci*, 2019). However, we did not find significant differences between estrus and non-estrus state mice on any test. Next, we

evaluated whether phase of estrus cycle impacted behavioral outcomes by including it as a factor in our statistical analysis, but again did not find a significant effect. This may be due to uneven distribution of cycle phases across groups on any given date, as we did not artificially sync cycles and prioritized testing all mice on the same date to minimize other experimental variables. We have added a sentence to the Results to reflect the apparent lack of effect of estrus on our behavioral measures. However, as we note in the Methods, most mice paused in or entered diestrus after the forced swim test (presumably due to the stress), and thus most brain tissue for sequencing was collected in the diestrus stage.

11) I am confused about the methods for RNA-seq (but maybe I missed something). The authors state first that male VTA and NAc samples were pooled 3 animals/sample, but then say that male and female VTA, NAc, and PFC samples were prepared from individual animals (4-6 independent samples/group).

We apologize for the confusion. The text should have read “male PFC and female VTA, NAc, and PFC” which we have corrected in the revised manuscript. Punches from 3 mice (similarly behaving, and within the same group) were pooled during RNA extraction for male VTA and NAc sequencing, which were performed first. Our initial reasoning was that pooling animals would increase RNA starting quantity for library prep, decrease sample-to-sample variability, and be more cost effective. However, as sequencing prices came down, and we were confident that we could generate high-quality libraries from punches from a single animal, we chose to retain “individual differences” and sequenced more samples/group for male PFC and all female samples.

Minor Comments:

1) While I believe that the behavioral replication cohort is important, the text associated with it could probably be majorly reduced. This text could instead be placed in the figure legend for Supplementary Figure S1.

We have reduced the main text and included the statistics in the Supplementary Figure S2 legend as suggested.

2) The text for Figure 5 axes is too small to read.

3) The heatmaps in Figure 2 seem kind of washed out.

We have increased the font size and embedded the heatmaps to hopefully improve color.

Reviewer #3 (Remarks to the Author):

In this paper, the authors extended their previous finding of the effect of early life stress on the response to social defeat stress in adult male mice, and associated transcriptional changes in ventral tegmental area (VTA). In this study, the authors performed similar experiments in female mice, and the effect of earlier stress was also examined in female mice. They also examined two brain areas, nucleus accumbens and prefrontal cortex, in addition to VTA. First, they performed behavioral analyses to show that there is an interaction between early life stress and stress in adults. Next, they performed RNAseq analyses and found that early life stress affected the stress-induced transcriptome changes in adults. They performed a number of bioinformatics analyses to interpret the obtained data.

Though the obtained results are robust and potentially interesting, the interpretation is sometimes arbitrary. No additional experiments to support the causal relationship between behavioral alterations and transcriptome changes were done, and thus this is a descriptive study. However, the experiments are well thought out, and the data are precious. The authors have made the data publically available, which will be useful for further studies.

The paper should be published after considering several minor issues listed below.

We appreciate Reviewer 3's recognition that, despite the lack of causal data, the data are “precious”

and useful for the field, and that the study should be published with minor revisions only. This underscores our point above under Reviewer 2 that causal data (which would have to be done in one sex/region/comparison at a time) would detract from the characterization of the extensive sequencing data presented together here.

1) Abstract. "associated with latent depression-like behavior". This is misleading because the interaction between early life stress and defeat was seen only for anxiety scores and the composite score of depression- and anxiety- related behavioral tests.

Our initial interpretation of the NSF test was as a measure of both anxiety-like and depression-like behavior, as reviewed by Samuels and Hen (2011) (and see below). However, we have removed the specific reference to latent depression-like behavior from the abstract.

2) Introduction. The authors state that early life stress (ELS) increases the risk of depression by two- to four- fold. This is overestimated. Retrospective assessment is affected by current mental status, and a prospective study reported lower odds ratio (Widom et al, Arch Gen Psychiatry 2007).

Thank you to the Reviewer for pointing out this prospective study. The odds ratios reported was 1.27-1.75 and this has been updated in the revised manuscript.

3) Results. The authors generated "a composite depression score". However, this includes the novelty-suppressed feeding test, which is a test for anxiety. Thus, the score should be named as "a composite depression and anxiety score".

We agree and have changed the wording to "composite behavior outcome score." Our interpretation of the NSF test is that it provides a measure of both anxiety- and depression-like behavior. In particular, it has predictive validity for amelioration of behavioral differences after chronic, but not acute, antidepressant treatment in rats and mice, which mirrors the effects of antidepressant treatment in human patients:

Bodnoff SR, Suranyi-Cadotte B, Aitken DH et al. (1988) The effects of chronic antidepressant treatment in an animal model of anxiety. *Psychopharmacology (Berl)* 95:298–302

Samuels B.A., Hen R. (2011) Novelty-Suppressed Feeding in the Mouse. In: Gould T. (eds) *Mood and Anxiety Related Phenotypes in Mice*. *Neuromethods*, vol 63. Humana Press

Santarelli L, Saxe M, Gross C et al. (2003) Requirement of hippocampal neurogenesis for the behavioral effects of antidepressants. *Science* 301:805–809

Surget A, Saxe M, Leman S et al. (2008) Drug-dependent requirement of hippocampal neurogenesis in a model of depression and of antidepressant reversal. *Biological Psychiatry* 64:293–301

David DJ, Samuels BA, Rainer Q et al. (2009) Neurogenesis-dependent and -independent effects of fluoxetine in an animal model of anxiety/depression. *Neuron* 62:479–493

However, we understand that other groups do not view the NSF test as reflecting depression-like behavior and therefore avoid the disagreement in the field by using the new term. We appreciate the Reviewer bringing this to our attention.

4) It seems that the use of a composite score was not intended at the initiation of this study, but it was used because there was no significant interaction for the results of the tests for depression-like behavior. If so, the authors should state that this analysis was done in the post hoc manner and has exploratory nature.

We actually did intend to use the composite score when the study was initiated. As noted above (Reviewer 2, comment 4), this analysis was done for consistency with our previously reported study of male depression-like behavior (Peña et al., *Science*, 2017). We have clarified this point in the methods description, in addition to the suggested sentence about its exploratory nature.

5) P4. The last sentence of 1st paragraph. The authors use the term. "transcriptional priming." This term is misleading. Observed difference of transcriptome may be caused by the structural differences such as neural morphology, cellular composition, etc. The term "transcriptional priming" gives

impression that the observed transcriptome difference is reflecting some intracellular molecular biological phenomena.

This is a fair point, and we have amended the text to reflect different possibilities for the observed transcriptional changes.

6) P7. Left column, Line 10."plasticity signature". This is too general, and misleading. Actually, it is the "transcriptome changes associated with *Lynx1* knockout".

We originally chose to use the same terminology as the original study from which this data were derived; nonetheless the Reviewer's point is well taken and we have amended the text to clarify that what is termed "plasticity signature" is a transcriptional signature of *Lynx1* knockout associated with cortical plasticity.

REVIEWERS' COMMENTS:

Reviewer #1 (Remarks to the Author):

The authors have adequately addressed my concerns.

Reviewer #2 (Remarks to the Author):

This is a resubmission of the manuscript, "Early life stress alters transcriptomic patterning across reward circuitry in male and female mice". The authors have been very responsive to my original comments. I appreciate that adding a manipulation might be too much for this manuscript, and think it is great that the authors validated a number of top hits using qPCR. Further, the authors directly addressed my other concerns in this revision. Together, I believe that this manuscript should be accepted for publication in Nature Communications.

Reviewer #3 (Remarks to the Author):

The authors adequately revised the manuscript in response to the comments by this referee. I have no additional comments.

Tadafumi Kato

Response to Reviewers NCOMMS-19-13687

Dear Reviewers,

Thank you very much to all of the reviewers for your time and thoughtful comments on this manuscript.

Catherine Peña and Eric Nestler (on behalf of all the authors)

REVIEWERS' COMMENTS:

Reviewer #1 (Remarks to the Author):

The authors have adequately addressed my concerns.

Reviewer #2 (Remarks to the Author):

This is a resubmission of the manuscript, “Early life stress alters transcriptomic patterning across reward circuitry in male and female mice”. The authors have been very responsive to my original comments. I appreciate that adding a manipulation might be too much for this manuscript, and think it is great that the authors validated a number of top hits using qPCR. Further, the authors directly addressed my other concerns in this revision. Together, I believe that this manuscript should be accepted for publication in Nature Communications.

Reviewer #3 (Remarks to the Author):

The authors adequately revised the manuscript in response to the comments by this referee. I have no additional comments.

Tadafumi Kato